# Tumor-Infiltrating B Lymphocytes: Promising Immunotherapeutic Targets for Primary Liver Cancer Treatment

**DOI:** 10.3390/cancers15072182

**Published:** 2023-04-06

**Authors:** Giulia Milardi, Ana Lleo

**Affiliations:** 1Hepatobiliary Immunopathology Labaratory, IRCCS Humanitas Research Hospital, 20089 Milan, Italy; 2Department of Biomedical Sciences, Humanitas University, 20072 Milan, Italy; 3Department of Gastroenterology, Division of Internal Medicine and Hepatology, IRCCS Humanitas Research Hospital, Rozzano, 20089 Milan, Italy

**Keywords:** tumor microenvironment, primary liver cancer, hepatocellular carcinoma, cholangiocarcinoma, B lymphocytes

## Abstract

**Simple Summary:**

Primary liver cancer is a frequent cause of cancer-related death with high mortality rates. Cellular components of the heterogeneous tumor microenvironment (TME) play a crucial role in promoting or inhibiting tumor growth. Indeed, a multitude of immune populations, including B cells, have been found within the liver TME. This review offers a comprehensive summary of B-cell biology and maturation process, as well as their phenotypic and complex functional properties in cancer. We also summarize the prognostic impact of B lymphocytes in liver malignancies and their potential benefit in the development of new immunotherapies for primary liver cancer treatment.

**Abstract:**

Hepatocellular carcinoma and cholangiocarcinoma are the fourth most lethal primary cancers worldwide. Therefore, there is an urgent need for therapeutic strategies, including immune cell targeting therapies. The heterogeneity of liver cancer is partially explained by the characteristics of the tumor microenvironment (TME), where adaptive and innate immune system cells are the main components. Pioneering studies of primary liver cancers revealed that tumor-infiltrating immune cells and their dynamic interaction with cancer cells significantly impacted carcinogenesis, playing an important role in cancer immune evasion and responses to immunotherapy treatment. In particular, B cells may play a prominent role and have a controversial function in the TME. In this work, we highlight the effect of B lymphocytes as tumor infiltrates in relation to primary liver cancers and their potential prognostic value. We also present the key pathways underlying B-cell interactions within the TME, as well as the way that a comprehensive characterization of B-cell biology can be exploited to develop novel immune-based therapeutic approaches.

## 1. Introduction

Primary liver cancer is considered the fourth most lethal malignancy worldwide [1], with hepatocellular carcinoma (HCC) and cholangiocarcinoma (CCA) being the two most common hepatic tumors. In 2020, 905,700 people were diagnosed with liver cancer and 830,200 of them died from it. Global age-standardized incidence and mortality rates were highest in Eastern Asia, Northern Africa, and South-Eastern Asia and increased in males more than females. Liver cancer was among the top three causes of cancer death in 46 countries and among the top five causes of cancer death in 90 countries [2]. Whereas HCC accounts for approximately 75–85% of the incidence of all liver cancers [3], CCA is rarer, but its incidence and mortality has been significantly increasing over the past decades, becoming a global health problem [4]. CCA accounts for about 10–15% of primary liver tumors and it is the second most common after HCC [5]. Primary liver cancer commonly arises under chronic inflammation and damage to the hepatic parenchyma or the biliary tract, which can originate from viral infections (hepatitis B virus (HBV) and HCV), metabolic alterations (alcoholic steatohepatitis (ASH), nonalcoholic steatohepatitis (NASH), chronic toxin exposure (aflatoxin), or parasite infection (flukes) [6,7]. Furthermore, lifestyle factors such as frequent alcohol consumption, diet, and sedentary lifestyles increase chronic inflammation and the incidence of hepatic tumors [7].

Currently, surgery is the only potentially curative treatment, sustained by other therapeutic options such as chemo-immune therapeutic agents (i.e., durvalumab, gemcitabine, and cisplatin for CCA and atezolizumab plus bevacizumab for HCC) or multikinase inhibitors (sorafenib and lenvatinib as first-line systemic therapy in HCC) [8], which provide only a limited extension of overall survival and a marginal increase of life quality [1,9,10,11,12,13]. The high recurrence rate after surgical resection, the refractoriness to systemic therapeutics observed in most of the patients, the low rate of molecularly targeted available therapies [14,15], and the inflammatory heterogeneity of liver cancer make the development of novel curative treatments harder. Therefore, there is an urgent need for valid therapeutic alternatives for patients affected by primary liver cancer, including immune cell targeting therapies. Immunotherapies approaches targeting checkpoint receptors expressed by tumor-infiltrating lymphocytes (TILs) have increased the overall survival (OS) of patients with multiple cancers [16,17]. TILs are relevant cellular components of the tumor microenvironment (TME) in liver malignancies (Figure 1), playing an important role in cancer immune evasion and responses to immunotherapy as immune checkpoint blockade (ICB) [18,19,20,21,22], but the value of most cellular components of the TME in the development and progression of primary liver cancer and drug resistance remains unknown. Thus, a comprehensive characterization of liver cancer heterogeneity, the architecture of the TME and its infiltration status could elucidate mechanisms of liver tumor progression to develop new and more effective therapeutic approaches. Recent studies demonstrated that a preponderance of CD8+ T cells within the tumor area and CD4+ T cells in the tumor–liver interface was positively correlated with OS [23,24], and then a high infiltration of hyperactivated CD4+ regulatory T cells within the tumor was associated with a worse prognosis [25,26], confirming their critical role in controlling tumor development. Many aspects of liver cancers related to T lymphocytes are undergoing extensive studies; contrarily, the role exerted by other immune cell components such as B lymphocytes in HCC and CCA needs a further developing analysis.

The purpose of this review was to provide insights into the biology of B lymphocytes in liver cancers such as HCC and CCA focusing on their distribution, molecular pathogenesis, prognosis-related importance, and potential for immunotherapy development.

## 2. Background of B Lymphocytes

### 2.1. B-Cell Biology and Maturation Process

B lymphocytes are one of the most important cellular components of adaptive immunity. They develop mainly in the liver during fetal life and in the bone marrow (BM) after birth, arising from hematopoietic stem cells (HSCs) [27] (Figure 2). B-cell development is regulated by intrinsic genetic programs and by external signals such as cytokines present in the microenvironments of the fetal liver and BM. BM stromal cells (including mesenchymal cells, osteoblasts, and endothelial cells) create a particular microenvironment, known as niches, that provide growth factors and cytokines crucial for the correct spatial distribution, survival, and differentiation of HSCs into mature blood cells [28]. The interaction between stromal cells and early B-cell progenitors gives rise to progenitors B (pro-B) that rearrange their Ig heavy chain genes to differentiate into precursor B (pre-B) cells [29]. Pre-B cells rearrange their Ig light chain genes and differentiate first into IgM+ immature B cells, then into IgM + IgD + mature B cells (termed transitional B) [30]. The rearrangement of B-cell receptor (BCR) genes is a unique mechanism to generate diversity in B cells from limited numbers of variable gene segments, through genetic recombination [27,31]. Up to 75% of immature B cells possess a nonfunctional or self-reactive BCR and undergo clonal deletion or receptor editing [31]. Transitional B cells that successfully produce functional and non-self-reactive BCR exit bone marrow, enter the blood, and migrate to secondary lymphoid organs (SLO) [32]. In the periphery, B cells continue their development and, since transitional B are short-lived cells, rapidly differentiate into mature-naïve B cells. These naïve B cells recirculate between blood and SLO, entering B-cell follicles in lymph nodes and the spleen, where they represent the majority of the B-cell pool and respond to antigen encounter with T-cell help, which finally leads to antibody production [33].

The cross-link between Toll-like receptors on the B cell’s surface and pathogen-associated molecular patterns (PAMPS) induce a B-cell activation that mainly occurs within the follicles of SLO [27,34]. Upon antigen recognition and activation, follicular B cells present their antigens to T-helper cells, to receive additional activation signals provided by T cells [35]. These B–T interactions promote the differentiation of activated-follicular B cells into rapidly dividing cellular components forming germinal centers (GC), different regions in secondary lymphoid organs composed of dividing B cells, T follicular helper (Tfh) cells, FDCs (follicular dendritic cells), and stroma cells. Herein, activated naïve B cells undergo a clonal expansion, introduce mutation in their immunoglobulin genes (class-switch recombination (CSR)), and change the specificities of their BCR by somatic hypermutation (SHM). Therefore, they are selected for their affinity to the antigen. Only high-affinity B cells develop into long-lived plasma cells (PCs) or switched memory B cells that express pathogen-specific antibodies (IgG, IgE, or IgA) [36,37].

A percentage of mature-naïve B cells follow an alternative T- and germinal-center-independent pathway and differentiate into short-lived antibody-secreting cells of lower affinity [38]. These short-lived plasma cells do not undergo affinity maturation or somatic hypermutation. However, they are crucial for the initial but not sustained response to antigens [38,39].

### 2.2. B-Cell Function in Adaptive Immune Response

The B-cell subsets, once mature and terminally differentiated into plasma and effector cells, are mainly involved in circulating antibodies secretion, thymic-independent IgM antibody responses and antigen presentation [40,41]. B lymphocytes are required for the initiation of T-cell immune responses, and recent studies have demonstrated that their absence impairs CD4+ T-cell priming [42]. The differential activation and expansion of CD4+ T cells by B lymphocytes could be associated with immune responses, making this a critical area for future studies of host defense and autoimmunity.

Importantly, interactions between B and T cells are based on the antigen presentation process as well as cytokines production. In the first case, the antigen needs to be internalized by the BCR, processed and then presented in an MHC-restricted manner to T cells [43,44]. As for the second case, B cells are capable of synthesizing and releasing several cytokines that exert a disease-causing/protecting effect on malignant tumors, infection, and autoimmunity [36,45]. In particular, B cells release both immunosuppressive and proinflammatory cytokines. Immunosuppressive cytokines include transforming growth factor (TGF)-β and interleukin (IL)-10 that can negatively regulate the immune response by suppressing T helper (Th) cell function, limiting the Th1 and Foxp3+ T regulatory cell (Treg) differentiation, by reducing antigen-presenting cell (APC) roles and proinflammatory cytokines released by monocytes, then causing CD4+ T cell death and CD8+ T cell anergy [46]. Inhibitory cytokines (such as TGF-β and IL-10) are also crucial in maintaining self-tolerance and immune homeostasis, with their positive role in shaping the immune system and the inflammatory responses [47].

On the other hand, B lymphocytes also produce positive immunoregulation factors including proinflammatory cytokines (e.g., IFN-α, IFN-γ, and TNF-α and IL-1 IL-2, IL-6, IL-8, IL-12, IL-16, and IL-35), Th2 cytokines (IL-13, IL-5 and IL-4), macrophage colony-stimulating factor (M-CSF), granulocyte-macrophage colony-activating factor (GM-CSF), hematopoietic growth factor granulocyte colony-stimulating factor (GCSF), and chemokines as CCL7 and CCL5 [36,48]. In particular, IL-2, 1L-4, IL-12, and IFN-α promote Th1, Th2, Th17 development and responses; GM-CSF triggers neutrophil response; IFN-α and TNF-α are involved in DC maturation and lymphoid configurations; IFN-α improves NK cells and macrophage activation, stimulating their own development, and promoting antibodies production [49].

B cells have a crucial role not only in immune system development, but also in its maintenance [50]. Overall, B lymphocytes release immunomodulatory cytokines that can influence T cell, DC differentiation, and APC functions. They can also regulate lymphoid tissue organization, neogenesis, wound healing, and transplanted tissue rejection and influence tumor growth as well as cancer immunity [24,51,52].

## 3. Effect of B Cells on Tumor

### 3.1. B-Cell Function in Tumor Condition

B cells are the second most abundant tumor-infiltrating lymphocytes (TILs) and, modulating the immune response, exert an important role in the adaptive immunity of cancer. They account for up to 25% of all cells in some tumors [53]. In the TME, tumor-infiltrating B lymphocytes (TIL-B) exhibit the typical hallmarks of B cells such as antigen recognition, clonal expansion, somatic hypermutation, class-switch recombination, and differentiation in antibody-producing plasma cells. Antibodies released by plasma cells can alter the function of their antigenic targets on cancer cells, opsonize tumor cells for the presentation and cross-presentation of tumor antigens by DCs, activate the complement cascade, or contribute to NK-cell-mediated tumor killing [53,54]. Antibodies against tumor antigens have been found in the serum of cancer patients, but their role in humoral immune responses against cancer development and progression remains controversial [55].

Moreover, TIL-B impact the adaptive immune responses (including CD4+ and CD8+ T cells) and innate mechanisms, involving complement, myeloid, and NK cells. B-cell effects on patient outcomes can be heterogenous, based on a specific tumor’s anatomic sites, histology, and molecular subgroup. The presence of infiltrating B lymphocytes can be associated with a positive prognostic value in most of the cancers, and with a negative outcome in others [18].

B-cell populations found in the TME of many cancer types are significantly heterogeneous in both the immunophenotype and functional role. Some authors demonstrated the presence of different phenotypes, including effectors and regularity B (Breg) cells that can exert both pro- or antitumor activities [31,56] (Figure 3). The dual role of B cells is influenced by several factors, such as hypoxia, cytokines and metabolites produced by other immune cells (e.g., Tregs and myeloid-derived suppressor cells, MDSC), and tumor cells [57].

TIL-B may act as APCs within the tumor tissue. They first recognize the antigens through BCR and present them to CD4+ T cells, sustaining the CD4+ and CD8+ T cells’ activation [58]. B cells, which exert the function of APCs, have often increased the expression of costimulatory proteins critical for T-cell activation such as CD80, CD86, and MHC-II proteins, suggesting the critical role of B cells in the strength and magnitude of the CD4+ and CD8+ T cell response in cancer [59].

The cytokine milieu is another crucial aspect in cancer progression or control. Tumor-infiltrating B-cell subsets secrete a variety of cytokines to regulate tumor immunity in a negative or positive manner. B lymphocytes can produce immunostimulatory cytokines (e.g., IL-6, IL-4, IL12p40, IL7, INF-γ, TNF, CCL3, IL-2, and colony-stimulating factor 2 CFS2 or GM-CSF) [60] and anti-inflammatory cytokines (IL-10, TGF-β, IL-35) [61] even within the tumor tissue [62].

### 3.2. Protumorigenic Function of B Lymphocytes

Several studies have revealed that cancer growth can be sustained by the tumor-promoting effects of Bregs and antibodies (Figure 3). Experiments carried out with multiple mouse cancer models have reported that a lack of B cells impairs the tumors’ growth, enhancing and accelerating cancer progression [63]. Indeed, B lymphocytes can exert their protumorigenic function in different ways: boosting the development of cancer cells by activating Fcγ receptors (FcγR) on myeloid cells, sustaining the generation of new blood vessels in tumors, producing lymphotoxin which activates cancer resisting castration, or modulating the signals of the IL-8/androgen receptor which increase tumor metastasis.

The tumor-enhancing capacity of B cells is mainly mediated by Bregs. Breg cells support carcinogenesis, tumor progression, and a metastatic process suppressing Th1, Th17, and CD8+ cytolytic T-cell responses, through the production of immunosuppressive cytokines (e.g., TGF-β, IL-35, and IL-10) [64,65]. IL-10 can inhibit other stimulating cytokines’ production, causing a decrease in the reactivity, not only of Th1, Th17, and CD8+ T cells, but also of NK cells [66]. The TGF-β production can drive the polarization of CD4+ T cells into active Tregs, inhibiting NK cells and CD8+ T cells, which are crucial for tumor growth inhibition [67]. Additionally, Bregs can promote the apoptosis of effector CD4+ T cells through the expression of Fas ligand (FasL) [52]. Furthermore, Bregs, releasing IL-10, induce the differentiation of tumor-associated macrophages (TAMs), skewed toward a M2 macrophage phenotype, that inhibits effector T and NK cells [56]. However, it is still unclear whether Bregs actively promote tumor growth, or whether an increase in the Breg population inhibits the immune response against tumors.

Antibodies released by mature B cells might contribute to tumor progression through the production of circulating immune complexes (CICs) [53]. A CIC is composed of antibodies bound to a soluble antigen and it may have a role in the setting of cancer. In human malignancies, CICs found in the peripheral blood or tumor tissue generally reflect poor clinical outcomes [68]. This protumorigenic role of B cells and CICs have been supported by studies using a genetic mouse model of squamous cell carcinoma [39], wherein CICs, present in premalignant tissue, induce chronic inflammation through their recognition and subsequent activation of FcγR on infiltrating myeloid cells, inducing a myeloid suppressor cell activity that promotes tumorigenesis [69].

Another important feature for cancer induction is lymphangiogenesis, which promotes tumor growth and metastatic process throughout the body. TIL-B may sustain tumor progression by providing lymphotoxin (a survival factor that can induce angiogenesis) and by promoting androgen-independent cancer progression by activating NF-κB and STAT3 pathways [53]. In particular, androgen ablation causes damage to the stromal cells of the TME, sustaining leukocyte infiltration into the tumor. Following androgen ablation, intratumor Tfh cells produce the chemokine CXCL13 that recruits B cells into the tumor through CXCR5 signaling [70,71,72]. These TIL-B then secrete lymphotoxin, which activates NF-κB signaling and STAT3 in the cancer cells, resulting in androgen-refractory growth and tumor progression [73,74].

Finally, B cells can also enhance metastasis by upregulating IL-8, which can modulate androgen receptor and matrix-metalloproteinase signaling [74].

### 3.3. Antitumorigenic Role of B Lymphocytes

Recent publications have showed a positive correlation between an increased frequency of CD20+ B cells and the clinical outcome of patients affected by melanoma, sarcoma, breast cancer, esophageal cancer, non-small cell lung cancer, colon cancer, or biliary tract cancer [72], suggesting that B cells may also exert an antitumor activity (Figure 3). TIL-B inhibit cancer development through the production of antitumor reactive antibodies and specific cytokines that coordinate other immune cells. B lymphocytes can enhance the cytotoxic activity of T cells, the phagocytosis by macrophages, the NK cells’ function of tumor killing, the tumoricidal effect by granzyme B secretion, and the CD4+ and CD8+ T cells’ priming. TIL-B seem to be involved even in the coordination and maintenance of tertiary lymphoid structures (TLSs) [54,73], particular cellular organizations very similar to lymph-node aggregates characterized by separated B- and T-cell areas, specialized populations of DCs, stromal cells, and high endothelial venules (HEVs); all of them are able to create cellular interactions similar to those that occur in secondary lymphoid organs [56]. TLSs develop de novo in inflamed nonlymphoid tissues and they are associated with chronic inflammatory disorders, autoimmunity, and cancer [54,74,75,76]. They usually arise in response to antigens and inflammatory stimuli [77].

B cells produce cytokines that can facilitate TLS formation at sites of chronic inflammation [78]. Several pieces of evidence indicate that TLSs play a major role in controlling tumor progression and they are related to better clinical outcomes in both human disease and mouse models [54,76]; for this reason, TLS structures are considered as prognostic markers predicting longer patient survival [79,80]. Thus, TLSs may represent the initiation of a local antitumor B-cell-mediated immunity. Considering the B-cell ability to release cytokines, B lymphocytes also recruit other immune cells to TLSs and effector sites and induce the antitumor cytolytic T-cell activity through the interaction between CD27-expressing B cells and CD70-expressing CD8+ T lymphocytes [81,82]. Tumor-associated TLSs may be the place where B cells undergo cell clonal amplification, somatic hypermutation, and class-switch recombination, revealing a local antigen-driven response and antibody affinity maturation [49,79]. Thus, B cells differentiate into plasma cells producing significant amounts of tumor-specific antibodies, or presenting tumor-derived peptides to T cells, modulating their phenotypes [54]. This system may be exploited clinically to improve patient prognosis and responses to immunotherapy.

In the long-term immune response between tumors and the immune system, the number of DCs become smaller and they can no longer sustain their role of presenting antigens; therefore, B cells act in the TME as local APCs and contribute to the survival and proliferation of tumor-infiltrating T cells [83]. TIL-CD20+ B have been found close to CD8+ T cells, and the presence of both CD20+ and CD8+ lymphocytes has been associated with longer survival especially in ovarian cancer, compared with patients affected by a tumor lacking B cells [84]. This suggests that B cells play the role exerted by APCs [85]. In murine models, CD4+ T-cell activation and clonal expansion in response to protein antigens and pathogens were impaired when an anti-CD20 monoclonal antibody (rituximab) was used for B cells’ depletion, also supporting here the critical role of B cells for an optimal antigen-specific CD4+ T-cell priming [86,87].

Moreover, B cells stimulated from tumor cells generate antitumor reactive antibodies, causing a strong humoral response. Tumor-specific antibodies (mainly IgG1 antibody class), secreted by plasma cells, can bind to FcγR and trigger the complement cascade, mediate the phagocytosis of tumor cells, the cytotoxicity of NK cells, and enhance antigen presentation by DCs [88,89]. Activated B cells also possess the tumor-killing potential and may directly destroy cancer cells by secreting TRAIL and granzyme B [90].

## 4. Clinical Application and Future Perspective of B Lymphocytes

B cells play an important role in cancer biology, exerting a protumorigenic or antitumorigenic effect. For this reason, researchers have considered them next-generation candidates for tumor immunotherapy. As mentioned before, B cells can be stimulated by tumor antigens and produce tumor-specific IgG-dependent antibodies [91], imprinting our body with a long-lasting immune memory. B lymphocytes and their subsets can also stimulate other components of the tumor-immune system, such as promoting Th1 cells, activating cytotoxic T-cells, and secreting cytokines [53]. Therefore, there are various categories of B-cell-based immunotherapies described below [92] (Figure 4).

Monoclonal antibodies (mAbs) are one of the most used curative treatments in support of conventional therapy such as radiation, chemotherapy, and surgery. For example, rituximab, an anti-CD20 able to deplete B cells, was applied for chronic lymphocytic leukemia (CLL) and B-cell lymphoma treatment but had limited success in solid tumors [93].

Treating patients affected by CLL with an anti-CD20 mAb (such as rituximab) can lead to the accumulation of Bregs and lymphoma-resistant cells. Since Bregs play a pivotal role in immune suppression, the depletion of this B-cell subset by using anti-IL10 antibody or chemicals products (resveratrol) appeared to be a promising solution for killing in vitro breast tumor cell lines. Moreover, preventing the conversion of naïve B cells into Bregs (using products such as lipoxins A4 and MK866) may also be an interesting strategy [66].

Cytokines released by Bregs are associated with tumorigenesis and their inhibition can be helpful for cancer treatment. As said before, Bregs secrete immunosuppressive cytokines, such as IL-10 and TGF-β. While IL-10 suppresses the function of cytotoxic cells (CD8+ T, NK, and Th1 cells), TGF-β promotes the differentiation of B cells into IgA plasma cells, which secrete IL-10 and express immunomodulatory receptors, such as PD-L1 and FAS-L, suppressing cytolytic activity. Consequently, specific drugs which induce B-cell depletion lead to a lack of Bregs production and mediate an antitumor activity [51,89].

B-cell activation can also have a crucial effect on tumor-growth suppression. Indeed, thanks to the CD40–CD40L costimulatory interaction, activated B cells cause the activation of cytotoxic T cells, which suppresses tumor growth. Thus, promoting B-cell activation and proliferation can be an important goal. Experiments have shown that the combined use of GM-CSF and IL-4 seems to induce B-cell activation and proliferation and be effective on cancer cell growth as melanoma cell lines [66].

Moreover, immunotherapy based on tumor-associated autoantibodies can also be used for therapeutic purposes. These autoantibodies have a variety of functions, such as antibody-dependent cellular cytotoxicity (ADCC), complement-dependent cytotoxicity (CDC), the cross-presentation of tumor antigens, and T-cell activation. A report suggested that P53 (a tumor suppressor protein) autoantibodies were associated with increased survival in HCC, while in other cancers, such as lung, colon, breast, and oral cancer, they were associated with poor survival [94].

Immunoglobulins are another resource important for therapy in tumor. They are secreted by both cancer cells and B cells. Cancer-derived immunoglobulins usually promote tumor growth by inducing inflammation and the activation of platelet aggregation and by escaping the infiltration of tumor cells. On the other hand, immunoglobulins derived from B cells have a tumor suppressor role. Importantly, this type of immunoglobulins is highly variable, since it is derived from somatic recombination during B-cell development. One of the most relevant seems to be the IgG class type, having a prognostic value in lung, colon, pancreatic, liver, gastric, ovarian, bladder, renal, salivary gland, soft-tissue, thyroid, and parathyroid cancers. Cancer-derived immunoglobulins can also be a useful resource, albeit with a limited diversity and less active function [89,92].

Recent studies have demonstrated a crucial association between B-cell-dependent antitumor immunity, TLS, and responsiveness to immune checkpoint blockade immunotherapy in different types of cancer [95,96]. TLSs are mainly composed of CD20+ B cells and CD8+ T cells, which infiltrate tumors, and their presence correlates with patient survival during immunotherapy. Some reports have validated them in metastatic melanoma [97] and sarcoma [98], but TLSs also have a prognostic role in breast cancer and colorectal cancer [99]. The presence of B cells and TLSs is strongly associated with a positive response to ICB in patients with soft-tissue sarcomas [98], metastatic melanoma [97], and renal cell carcinoma [100]. TLS gene expression signature was able to predict clinical outcomes to ICB with anti-CTLA-4 and/or anti-PD-1 in melanoma samples [98]. Even different maturation stages of TLSs were related to the clinical outcome of patients; TLS creation and reduction in GC development blunted antitumor immune response [101,102].

High-profile studies from human patients and mouse models have demonstrated that TIL-B, present in TLSs, are important players in responses to immunotherapies and outcomes of cancer patients [103,104,105]. This happens because inside mature TLSs, B-cell clones are selectively activated and amplified and acquire an activated/memory phenotype, expressing markers of antigen presentation (MHC class I and II) and costimulatory molecules such as CD40, CD80, CD86, and PD-L1; thus, they become easily affected by anti-CTLA-4 and anti-PD-1/PD-L1 ICB [53,103]. It is still unclear whether TLS frequency increases response to ICB treatment, but a histology analysis described that the CD20 density was higher at the baseline for immunotherapy-responding patients and increased after ICB treatment, while nonresponding patients had a low CD20 density before and after therapy [5]. Further studies will help to establish whether ICB administration actively induces TLS formation, and if B lymphocytes and TLS creation have an active and beneficial role in immunotherapy response. In this way, clinical strategies based on drugs that modulate B lymphocytes or TLS formation have been studied extensively to fight solid tumors [102,106,107].

## 5. Effect of B Cells on Primary Liver Tumors

### 5.1. Background of HCC

Hepatocellular carcinoma is considered the major subtype of liver cancer, the sixth most common form of cancer worldwide, and the third most significant factor of cancer mortality [108]. HCC frequently arises when the liver is affected by chronic disease and cirrhosis [109]. Indeed, the common risk factors for HCC include chronic infection with HBV, HCV, alcohol abuse, and exposure to aflatoxin B1, as well as smoking, obesity, NASH, diabetes, inherited disorders such as hereditary hemochromatosis, and adenomas [100,101]. HCC is often diagnosed in an advanced stage, when the only treatment includes surgery, combined with radio and chemotherapy. However, the rate of recurrence in HCC patients is high, and the death rates increase by 2–3% per year [101].

HCC is a highly heterogeneous malignancy. This heterogeneity is reflected in the disease progression and treatment of the individual patient and it is closely related to the TME, which comprises cellular and noncellular components [102]. The major cellular components include tumor cells, activated hepatic stellate cells, MDSCs, cancer-associated fibroblasts (CAFs), tumor-associated macrophages (TAMs), tumor-associated neutrophils (TANs), immune and endothelial cells [103]. The tumor stroma is considered the noncellular component and includes extracellular-matrix (ECM) proteins, proteolytic enzymes, cytokines, and growth factors [104].

Based on tumor-infiltrating immune cells, Zhang et al. classified HCC samples into three immune subtypes: “immunocompetent”, “immunosuppressive”, and “immunodeficient” subtypes [110]. The immunocompetent subtype was characterized as CD45^high^ FOXP3^low^, with normal T-cell infiltration. On the other hand, the immunosuppressive subtype showed high frequencies of immunosuppressive cells (as regulatory T and B cells and immunosuppressive macrophages) and a higher expression of immunosuppressive molecules (PD-1, PD-L1, TIM-3, CTLA-4, VEGF, TGFβ, and IL-10). Finally, the immunodeficient subtype exhibited a lower infiltration of lymphocytes [111,112].

Current studies have demonstrated a marked heterogeneity in HCC tumors and highlighted that the crosstalk between cancer cells and the liver microenvironment promotes tumorigenesis and HCC pathogenesis, by pushing cell proliferation, survival, and the ability of migration and evasion [108]. In detail, the TME in HCC appears to be immunosuppressive, promoting the proliferation, invasion, and metastasis of tumors [106,107]. Thus, a better characterization of the immune landscape of HCC at high resolution would facilitate refining patient stratification and identify putative immune-therapeutic targets in order to develop novel therapies for HCC. For this reason, targeting the TME with new immunotherapy strategies (including vaccines, antibodies, immune checkpoint inhibitors, and adoptive cell therapy (ACT)) is the pivotal goal for HCC treatment [107]. Furthermore, the immunological classification could guide immunotherapy for HCC, thus deciding to administer an agonist antibody for the immunosuppressive subtype or immune checkpoint inhibitors (as PD-1 blocking antibody) for the immunoreactive subtype [113].

The research on the biological characteristics of T cells is exponentially increasing but gaining more insights into B-cell biology related to liver HCC cancer is needed. Indeed, B cells infiltrate the HCC tissue becoming an active component of the TME and they can be identified at various phases of HCC development [36]. Importantly, their presence may be different according to the stage and histology-related subcategory [36]. The knowledge of B-cell biology related to cancer is not complete, but we know B lymphocytes impact even the humoral and cellular immunization of HCC.

### 5.2. B Lymphocytes in HCC

B cells are considered an important part of the TME, but the distributing process, frequency, and prognosis-related importance exhibited by invasive B-cell subsets within HCC is still controversial [50]. So far, it has been speculated that CD20+ B cells may have a dual effect on HCC [114]. Previous studies have revealed that CD20+ B cells can promote Tregs proliferation, suggesting a correlation between B cells and a worse overall survival and recurrence-free survival (RFS) [115]. Contrarily, Li et al. demonstrated that an increased CD20+ infiltration, supported by a high frequency of CD8+ T cells, was related to a decreased infiltration of Foxp3+ Tregs and CD68+ macrophages [116]. Immunofluorescence and flow cytometry analyses performed on HCC tissues showed that all B-cell subpopulations generally were reduced in tumors, with respect to the tumor-free liver area [117]. In detail, B-naïve (BN) cells and switched memory B (SM B) cells were considered prognostic factors for HCC survival, and, if highly concentrated within the tumor tissue, they led to more effective clinical results [117]. However, another study reported that the presence of TIL-B in human HCC was related to an increased tumor invasiveness and a decreased disease-free survival [118]. All these data suggest that multiple B-cell subpopulations coexist in the TME of HCC and exert dual effects on the tumor.

As for the antitumor function, in HCC, BN and SM B cells are positively correlated with a higher survival rate [36,119]. Indeed, BN cells, characterized by a higher expression of CD80, CD86, CXCR3, CCR5, and PD1, help the T-cell activation, which is also essential for HCC tumor control. In addition, SM B cells can produce IgG or IgA to promote humoral immunity. In brief, B cells infiltrate HCC tumor tissue and thus exhibit an anticancer effect at different stages [83].

Experiments carried out with mouse models showed that B-cell depletion enhanced HCC tumor growth and decreased local T-cell activation [86]. In humans, researchers highlighted that a high infiltration of CD20+ B cells caused the prolonged survival of patients. B lymphocytes may produce antitumor effects in different ways: they secrete specific antibodies to directly interact with tumor cells and exert humoral immunity; B cells can also act as an alternative APCs promoting CD4+ and CD8+ T cells’ response; they stimulate NK cells to directly kill tumor cells and release proinflammatory factors involved in T-cell activation [119]. According to Shi et al., TIL-B of HCC produce high levels of IFN-γ and IL-12p40, stimulating the CD8+ T cell response and exerting a direct killing effect [36,89].

On the other hand, the protumorigenic role in HCC cancer type may be exerted by Bregs, which exercise an inhibitory effect on the immune system by upregulating the expression of genes involved in tumorigenesis or by reducing the immune response. In this way, they can inhibit cytotoxic CD8+ T cells, the inflammation of Th1 cells and Th17 cells, facilitating the differentiation of Treg cells [114]. In human HCC, the phenotype, function, and clinic-related relevance of Breg cells have rarely been investigated, but whether or not the percentage of Breg cells is significantly higher within the tumor, especially in the late stage, they directly interact with HCC cancer cells through CD40/CD154 signaling and promote the growth and invasion of this malignancy [116]. Bregs secrete immunosuppressive cytokines to promote cancer, inhibiting T-cell function. They operate through an IL10-dependent pathway to induce T-cell dysfunction, creating conditions that lead to tumor progression [117]. Moreover, Ouyang et al. reported that more than 50% of infiltrating B cells in HCC were characterized by a low/activated phenotype of FcγR, and the high infiltration of these types of cells was positively correlated with cancer progression [118].

### 5.3. Background of CCA

Cholangiocarcinoma is a rarer and heterogeneous type of cancer originating at any point of the biliary tree, from the intra- and extrahepatic regions, and it is the second most common liver cancer accounting for 10–15% of all primary hepatobiliary malignancies [120,121]. Based on the different anatomical regions where the tumor grows, we can describe three different CCA subtypes: intrahepatic CCA (iCCA), perihilar CCA (pCCA) and distal CCA (dCCA) [121,122]. Different risk factors seem to lead to CCA development: liver flukes (*Opistorchis viverrini* and *Clonorchis sinensis*), primary sclerosing cholangitis (PSC), viral hepatitis (HBV and HCV), even metabolic syndrome, alcohol, and smoking [123,124]. Its incidence and mortality rate have been significantly increasing in Europe and North America over the past decades, due to the aggressive evolution of the disease and the lack of efficient diagnostic and therapeutic treatments [3,116]. CCA is very often diagnosed in an advanced stage, when potentially curative treatments, such as surgical resection, cannot be applied [124]. Indeed, the patient’s prognosis has not improved substantially, due to the high recurrence rate after surgical resection, the refractoriness to systemic therapeutics observed in most the patients, the low rate of molecularly targeted available therapies [125], and the lack of prognostic and predictive biomarkers. However, 13–14% of CCA patients showed a remarkable response to immune checkpoint inhibitors with a stabilization of the disease, demonstrating that immune-targeted therapies could be achievable for at least a subset of patients [126,127,128]. Why some patients respond to immunotherapy, whereas others do not, remains unclear.

Current research implies that the CCA phenotype is determined by genetic and epigenetic alterations in the cancer cells, by the molecular crosstalk between malignant cells and the components of the TME [129]. Based on the specific TME, CCA can be categorized depending on the presence or lack of immune cell infiltration. Overall, CCA is poorly infiltrated by the immune system, and is generally defined as a “cold” tumor; otherwise, the tumor infiltrated by lymphocytes is called “hot” [23]. The TME includes diverse populations of immune cells, e.g., T cells, B cells, myeloid lineage leukocytes, NK cells, macrophages and/or dendritic cells, that contribute to pro- or antitumor activities. Tumor-infiltrating lymphocytes (such as T cells, B cells, and NK cells) are the most involved in the immune response against tumor cells, and they are also responsible for the development of antitumor immune responses [130]. What we know is that the superior responsiveness to immune checkpoint inhibitors (such as anti-PD-1) is mediated by the reactivity of expanded CD8+ T cells towards the tumor antigens [131,132], especially in tumors with a high mutational burden, with mismatch repair deficiency and a high DNA microsatellite instability, typical of CCA [133,134].

As mentioned before, little is known about the nature, phenotype, and functions of other tumor-infiltrating lymphocytes such as B cells and their subsets in primary liver tumors, especially in CCA. However, future perspectives should pursue this direction: understanding B-cell molecular mechanisms involved in CCA progression or control will help to develop new potential target therapies.

### 5.4. B Lymphocytes in CCA

In contrast to T cells, B lymphocytes have been poorly examined in CCA and less evidence is available. Phenotypic and molecular studies carried out in CCA patients have shown that the frequency of CD20+ B cells in the TME is lower with respect to CD8+ and CD4+ T lymphocytes that represent the majority of TILs [26,135]. In one study, Kasper et al. demonstrated that CD20+ cells infiltrated more in the peritumoral area than in the tumoral site [23], but the transcriptomic landscape of B cells from tumoral and peritumoral tissues looked very similar based on gene expression profiles [26]. However, further studies are needed to explore their specific location under different situations.

In a cohort of 308 CCA patients affected by high-level microsatellite instability (MSI-H), Goeppert et al. discovered that higher numbers of CD8+ T cells and CD20+ B cells were related to a longer overall survival [134]. Current single-cell sequencing data of CCA demonstrated high B-cell infiltration levels in the TME were positively correlated with the prognosis [134]. Unfortunately, due to the limited number of studies, the prognostic role of B cells is not conclusive and more related research is needed to unravel their defined impact on long-term outcome.

TLSs are associated with favorable prognoses in several cancers, but their role in cholangiocarcinoma remains unclear. A comprehensive evaluation of the spatial distribution, abundance, and cellular composition of TLSs was performed in iCCA patients and revealed the opposite prognostic impacts of TLSs located within or outside the tumor. This suggests that the difference may be mediated by the different immune cell subsets present within the TLSs [135].

### 5.5. B-Cell-Based Immunotherapeutic Strategies in Primary Liver Cancer

Regulatory cells, including Tregs and Bregs, participate in monitoring internal immune homeostasis, and they have inhibitory roles in antitumor immunity in liver cancer. Innovative strategies based on B cells’ value are mainly focused on negative lymphoid regulatory cell blockage and B-lymphocyte targeted strategies [136,137].

In the first case, the strategy aims to block regulatory cells that mediate immunosuppression by depleting effector regulatory cells or modulating activating pathways. This can be a solution to achieve immunotherapy against liver cancers.

Secondly, therapeutic strategies, which correct dysregulations in B cells, are likely to generate beneficial antitumor immunity [138]. For example, clinical studies carried out on patients with type II diabetes have shown a high frequency of immature/transitional B cells, which might be liable for the progression of chronic hepatitis C (CHC) to HCC and are considered a potential disease predictor for CHC [139]. The correlation between B cell dysregulations, metabolic changes, or subset redistributions and the tumorigenesis of primary liver cancers is still less clear.

Experiments performed in Mdr2−/− mice under liver fibrosis condition showed that the depletion of CD20+ B cells with CD4+/CD8+ T induced inhibitory effects on liver cancer progression [139], while clinical studies in human patients revealed that B cells were notably decreased in HCC, and the density of tumor-infiltrating CD20+ B cells was positively correlated with superior survival [19,83]. Therefore, the deletion of specific B-cell subsets could be a solution.

A comprehensive characterization of molecular interaction between TIL-B and other cellular components of the liver TME is needed to improve the immunotherapy’s efficacy. For example, CD40L/CD40 interaction is a molecular mechanism engaged in the immune system activation. CD40, a member of TNF receptors, is expressed on the surface of cells such as DCs, monocytes, B cells, and some tumor cells [140]. The agonistic reagents to CD40 are promising immunotherapeutic candidates, since they have showed activation impacts on antitumor immunity. Indeed, mAbs that bind CD40 activate DC, myeloid cells, as well as B cells and increase their ability to process and present tumor-associated antigens (TAA) to local cytotoxic T lymphocytes [141,142]. The combination of anti-CD40/PD-1 with chemotherapy significantly impaired tumor growth and prolonged survival in advanced iCCA murine models [137]. A clinical trial to evaluate the efficiency and tolerability of an agonist CD40 antibody (CDX-1140) in advanced malignancies including primary liver cancers has recently commenced and it is recruiting for the next-step estimation [143].

A novel PD-1^hi^ B-cell subset, with a protumorigenic role, was identified in the TME of patients with HCC, whose frequency correlated with the disease stage and was associated with an early recurrence of HCC [144]. PD-1 ^hi^ B cells suppress tumor-specific T-cell response via IL10-dependent pathways upon interacting with PD-L1 to cause T-cell dysfunction and disease progression. Therefore, targeting immune checkpoint inhibitors and blocking the PD1/PDL1 axis through antibodies or miRNAs is a promising therapeutic strategy, since they may enhance immune cell antitumor responses [145]. Anti-PD-1 or anti-PD-L1 antibodies may block not only the PD-1 co-inhibitory pathway in T cells, but may also abolish the suppressive function of regulatory B cells [146].

TIGIT is another inhibitory receptor usually expressed on Tregs, that contributes to their suppressor function by limiting proinflammatory Th1 and Th17 responses. Indeed, TIGIT directly acts on CD155 expressed on activated T cells, resulting in the suppression of the T-cell response. It has been found significantly expressed in memory B cells, with the capacity to limit CD4+ T-cell proliferation and Th1/Th17 activation. Neutralizing TIGIT with anti-TIGIT antibodies has resulted in the partial recoveries of IFNγ and IL-17 expression by CD4+ T cells, suggesting it is a considerable marker to reactivate the immune system against tumor development [146].

Considering that cellular organized aggregates (such as TLSs) found in the TME of primary liver cancer can influence carcinoma occurrence and immunotherapy efficacy, the identification and modulation of TLSs is another powerful weapon against cancer in clinical practice. TLSs have been considered a marker of immunotherapy able to predict its effect and help to identify patients who respond to immunotherapeutic treatment [147,148,149]. Modulating TLS formation using chemokines/cytokines, immunotherapy, or the induction of the high endothelial vein has been studied extensively in primary liver cancer, in order to interfere with tumor growth and solve the problem of the low response rate to ICB therapy [149].

## 6. Conclusions and Perspectives

Our knowledge of the origin and detailed molecular characterization of primary liver cancer has progressed in the past years, but unfortunately, treatment options that confer to patients a longer survival and improvement in their quality of life have remained scarce [109]. A better understanding of the phenotype and molecular landscape of liver TME components could allow for an enhanced lymphocyte infiltration and effector function inside the tumors. In this way, it would be easy to target and reshape the hepatic inflammatory and/or metabolic microenvironment, to re-establish an effective immunosurveillance and response to immune checkpoint blockade [150].

B cells are an important component of the TME of various solid tumors, including primary liver cancers, and play different roles in the immune system. They have a contradictory value in tumor development: TIL-B can limit tumor growth by secreting immunoglobulins, boosting T-cell response, and directly killing cancer cells; then, they are also involved in the generation and maintenance of TLSs, promoting TILs infiltration into the tumor [151]. B-cell populations within the TLS and B-cell-related pathways mainly contribute to an antitumor response, improving patient outcomes. On the other hand, an increased frequency of a regulatory B-cell subset can facilitate tumor growth by secreting immunosuppressive cytokines including IL10 and/or TGF-β [51].

The complex distribution, multiple functions (pro- and antitumorigenic), and the presence of different B cell subsets in the TME, especially in liver cancer types, make it challenging to identify optimal targets for developing novel immunotherapy approaches.

In HCC, TIL-B mainly have an anticancer effect, but the immunoregulation function of Bregs cannot be ignored [152]. Further research on Bregs should be performed since they may represent a great potential for the development of new therapeutic strategies for hepatic malignancies. At different stages of HCC development, diverse subgroups of B cells play different roles, which are determined by the changes in the TME. Overall, the complex relationship among B cells with the TME of HCC and CCA remains unclear, but novel breakthroughs may be performed in the future.

Even though liver cancer such as CCA is poorly infiltrated, and B cells represent a small portion of TILs, a high frequency of B lymphocytes seems to be present in some CCA patients and has been related to longer OS and RFS. This might be attributed to a local immunological activation that leads to an increased number of cytotoxic CD8+ T cells [4,135]. However, given the limited quantity of quality data on B cells in CCA patients, further research is warranted to draw valid evidence-based conclusions.

The advent of immune checkpoint blockade in combination with radiotherapy and chemotherapeutic drugs has changed the treatment for multiple types of cancer, including liver cancer such as HCC, by extended survival times for a subset of patients. An immunosuppressive TME, typical of HCC, allows cancer cells to escape destruction by the immune components and develop resistance to immunotherapy [153]. Consequently, the development of novel targeted therapies can be helpful. The research focused on innovative immunotherapeutic approaches based on B-cell reprogramming/deletion is still in progress, but more studies on B-cell molecular features in primary liver cancers and associated mechanisms with the TME are needed to reach the next clinical stage. Considering the dual role of B lymphocytes within the tumor tissue, the new therapeutic strategies include specific monoclonal antibodies or chemical products with the main goal of depleting TIL-B subsets and blocking the regulatory cell response whether B cells are associated with worse clinical outcome, or targeting B lymphocytes to boost their APC capacity and the direct tumor killing effect if their high frequency is related to a better OS.

Moreover, since CD20+ B cells seem to be well aggregated and structurally organized in the TME of liver cancers, modulating TLS development in the hepatic tumor milieu may be a promising strategy in cancer treatment, as well as extend the use of TLSs and B cells as prognostic/predictive biomarkers for the ICB response.

However, a rigorous and deeper phenotypical and molecular analysis of B-cell subtypes in HCC and CCA is required to elucidate the effective role of B cells in tumor progression or control, to adjust the responsiveness to the immunotherapy that is already in the clinical stage and to develop new ones.

## Figures and Tables

**Figure 1 cancers-15-02182-f001:**
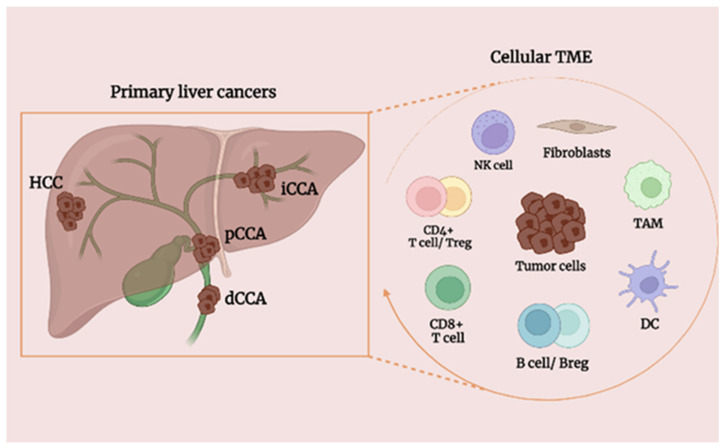
Schematic representation of the anatomic site of primary liver cancers and the cellular components of the tumor microenvironment (TME). Abbreviations: HCC: hepatocellular carcinoma; iCCA: intrahepatic cholangiocarcinoma; pCCA: perihilar cholangiocarcinoma; dCCA: distal cholangiocarcinoma; DC: dendritic cell; NK: natural killer cell; TAM: tumor-associated macrophage; Breg: B regularity cell; Treg: T regularity cell.

**Figure 2 cancers-15-02182-f002:**
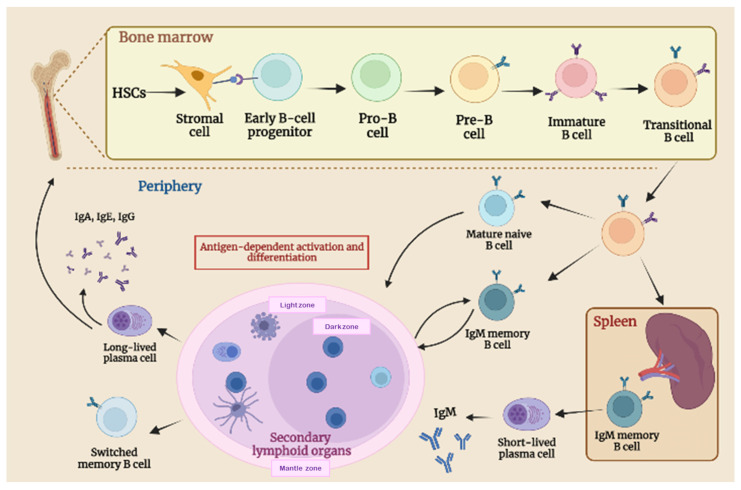
Schematic representation of B cell development and maturation stages. B cells develop in the bone marrow (BM) from hematopoietic stem cells (HSCs), progressing from early B-cell progenitors to circulating transitional B. Upon antigen recognition, transitional B cells become activated-naïve B cells that migrate to secondary lymphoid organs and enter germinal centers, where they undergo a clonal expansion and somatic hypermutation (SHM). Only functional B cells with a high receptor affinity interact with follicular dendritic cells (FDCs) and T follicular helper cells (Tfh), undergo a class-switch recombination (CSR), and differentiate into switched memory B cells and long-lived plasma cells (PCs), the final stage of B-cell development. As part of the extrafollicular response, mature B cells differentiate into short-lived plasma cells.

**Figure 3 cancers-15-02182-f003:**
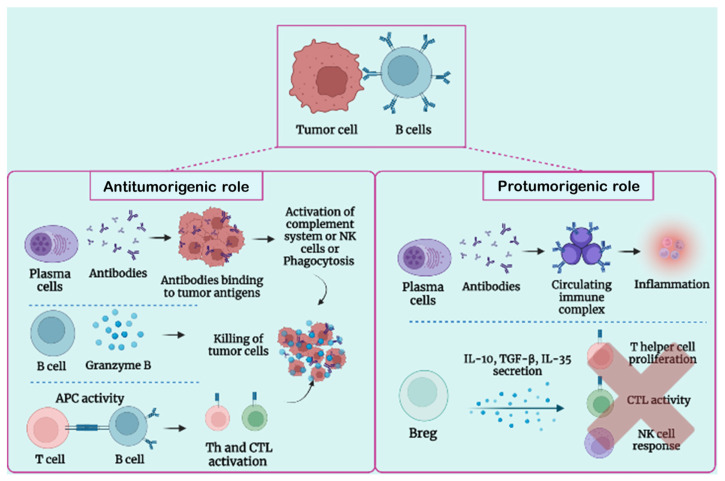
Immunological characterization of the dual B-cell role in cancer. Schematic representation of the tumor growth control sustained by effector B cells and antibodies binding tumor-associated antigens (left part of the scheme) in comparison with the circulating immune complex and regulatory B cells implicated in the cancer growth (right part of the figure).

**Figure 4 cancers-15-02182-f004:**
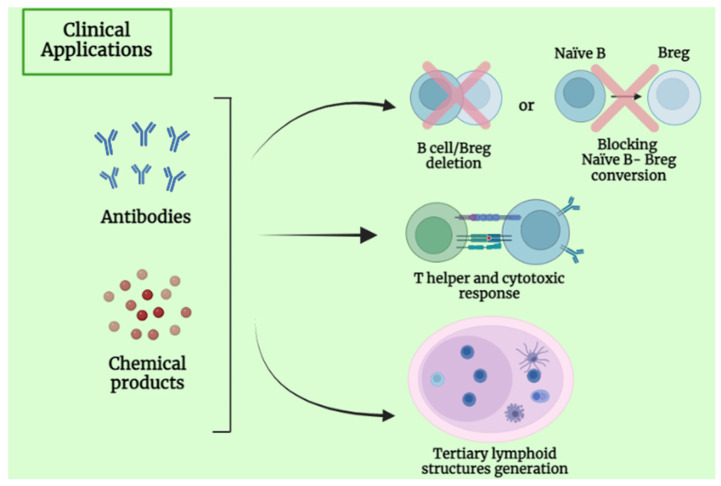
Immunotherapeutic strategies for solid tumors based on B-cell target and B-cell reprogramming, using monoclonal antibodies or chemical products.

## Data Availability

Not applicable.

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
