# Peer review of "Tumor-Infiltrating B Lymphocytes: Promising Immunotherapeutic Targets for Primary Liver Cancer Treatment"

_cancers, 2023, doi:10.3390/cancers15072182_

Round 1

Reviewer 1 Report

The authors have put together a detailed and well-referenced review on B cell biology, the role of B cells in cancer, and specifically in primary liver cancers. While other reviews exist on this topic this paper offers additional information and can contribute to the collection of papers. A few items should be addressed to improve the quality and readability of the paper.

1.     Typos – there are many typos. The authors need to edit the entire paper. Misspellings, misuse of tense, and improper grammar are prevalent and affect the readability of this review. A few important examples are included below but others need to be fixed as well.

a.     The abstract has several repeated words, e.g. “...immune system cells are main key components”

b.     In the introduction “.... mortality are significantly increasing over the past decades...”

c.     Section 3.1 3rd paragraph “regularity B (Bregs) cells...”

2.     Figure 2 appears to show IgM on the Pro-B cell

3.     The authors state that “...IL-10 that can negatively regulate the immune response by... limiting Foxp3+ regulatory T cell differentiation,...” This needs to be clarified

4.     Section 3.2 the authors state that “ ...a lack of B cells impairs the tumors’ growth, enhancing and accelerating the cancer progression.”

5.     Section 3.2 end of the first paragraph. The last 4 sentences have no reference.

6.     There are two sections on TLSs that are separated. These need to be reorganized.

7.     HCC infiltrating Bregs express high levels of PD-1. Checkpoint blockade of the PD1/PDL1 axis needs to be addressed by the authors. The authors should also consider discussing other targetable checkpoints on B cells, e.g TIGIT.

Reviewer 2 Report

My comments:

 1.      This review is quite comprehensive and updated. But the authors should also mention that, from the histological point of view, actually majority of HCCs do not show lymphocytes or inflammatory cell infiltration at all. Thus, many studies on the TIL reaction before or after treatment might be all from animal models. CCA is also similar. The center of CCA tumor often has a fibrotic center with sparce tumor cells and no inflammatory cell at all. So, it is hard to imagine how those published studies evaluated the role of inflammatory cell to cancer growth or anti-cancer. It is also why lymphoepithelioma-like HCC or CCA became an interesting subtype for research, since they typically have heavy lymphocyte infiltration.

2.      The authors have used a lot of abbreviations, which it is very difficult to remember. A detailed abbreviation list should be made for the readers.

Reviewer 3 Report

The authors reviewed the dual roles of pro-tumorigenic and anti-tumorigenic effects of tumor-infiltrating B lymphocytes (TIL-B) after a comprehensive summary of B cell biology and maturation process, then focused on TIL-B in hepatocellular carcinoma and cholangiocarcinoma. Though TIL-B are less concerned in primary liver cancers, compared to T lymphocytes, this paper is logically structured and provides valuable perspectives of new immunotherapies based on TIL-B. There are two minor suggestions for authors: 1. Please update the epidemiological data of primary liver cancers, such as the ranking of mortality and proportion of HCC in all liver cancers, according to Global Cancer Statistics 2020 issued by WHO. 2. Multikinase inhibitors such as sorafenib and lenvatinib are still the first-line systemic therapy, though not the preferred regimens.

Reviewer 4 Report

Hepatocellular carcinoma and cholangiocarcinoma are the fourth most lethal primary cancers worldwide. The authors offered a comprehensive summary of B cell biology and maturation process, as well as their phenotypic and complex functional properties in cancer. They also summarized the prognostic impact of B lymphocytes in liver malignancies and their potential benefit in the development of new immunotherapies for primary liver cancer treatment. The review is very interesting , however, this article has many problems:

1.    The authors should systematically introduce the immunological characterization and the immunoclassification of hepatocellular carcinoma.

2.    The author had better add a summative figure about the effect of B cells on primary liver tumors and the future perspective of B cell-based immunotherapeutic strategies in primary liver cancer.

Reviewer 5 Report

The review by Miladi and Lleo proposes to address and highlight the  potential of targeting Tumor infiltrating B cells as therapeutic targets of treating liver cancer. Unfortunately, the review falls short of achieving that. The major portion of the review is spent on explaining the development  of B cells, role in different arms of immune responses and role in general tumor biology. The review addresses the actual topic - which is liver cancer, as per the title of the article - only in the very last portion of the manuscript. 

Round 2

Reviewer 4 Report

  • The authors have revised the paper well as required.
  •  
  •  

Reviewer 5 Report

The authors have made changes to the manuscript as suggested. 

The article can be accepted for publication in Cancers.